# Plant dieback under exceptional drought driven by elevation, not by plant traits, in Big Bend National Park, Texas, USA

Elizabeth F. Waring and Dylan W. Schwilk

Texas Tech University, Department of Biological Sciences, Lubbock, TX, USA

## ABSTRACT

In 2011, Big Bend National Park, Texas, USA, experienced the most severe single year drought in its recorded history, resulting in significant plant mortality. We used this event to test how perennial plant response to drought varied across elevation, plant growth form and leaf traits. In October 2010 and October 2011, we measured plant cover by species at six evenly-spaced elevations ranging from Chihuahuan desert (666 m) to oak forest in the Chisos mountains (1,920 m). We asked the following questions: what was the relationship between elevation and stem dieback and did susceptibility to drought differ among functional groups or by leaf traits? In 2010, pre-drought, we measured leaf mass per area (LMA) on each species. In 2011, the percent of canopy dieback for each individual was visually estimated. Living canopy cover decreased significantly after the drought of 2011 and dieback decreased with elevation. There was no relationship between LMA and dieback within elevations. The negative relationship between proportional dieback and elevation was consistent in shrub and succulent species, which were the most common growth forms across elevations, indicating that dieback was largely driven by elevation and not by species traits. Growth form turnover did not influence canopy dieback; differences in canopy cover and proportional dieback among elevations were driven primarily by differences in drought severity. These results indicate that the 2011 drought in Big Bend National Park had a large effect on communities at all elevations with average dieback for all woody plants ranging from 8% dieback at the highest elevation to 83% dieback at lowest elevations.

## INTRODUCTION

The study of plant community structure over elevational gradients has played an important role in plant ecology (*Merriam, 1890*; *Whittaker & Niering, 1965*; *Whittaker & Niering, 1964*; *Whittaker & Niering, 1968*). In arid environments, water availability can increase dramatically with elevation and can drive turnover in plant species over short geographical distances (*Whittaker & Niering, 1968*; *Allen, Peet & Baker, 1991*). Drought can result in plant mortality or in stem death and partial dieback (*Sperry & Ikeda, 1997*; *Jacobsen et al., 2008*) and can lead to changes in the plant community structure

Corresponding author
Dylan W. Schwilk,
dylan.schwilk@ttu.edu

(*Lloret, Siscart & Dalmases, 2004*). Although precipitation in arid and semi-arid ecosystems is highly variable from year to year (*Noy-Meir, 1973*; *Schwinning & Sala, 2004*), a major decrease in annual precipitation could cause dramatic mortality or dieback for species near the limit of their ecological tolerance (*Pockman & Sperry, 2000*).

The Chisos Mountains of Big Bend National Park in southwest Texas have dense oak-juniper forest communities in the higher elevations while the lower elevation Chihuahuan desert communities are dominated by grasses, succulents, and drought-tolerant deciduous shrubs. The various plant communities in Big Bend National Park would be expected to respond to environmental stress differently due to different species morphology and physiology. Leaf traits can vary greatly among species and across plant communities. A leaf trait commonly used to measure energy invested per photosynthetic return is leaf mass per area (LMA, g/cm$^2$). Adaptations to prevent water loss, such as fibrous content and pubescence, will cause a thickening of the leaf, increasing its LMA (*Monneveux & Belhassen, 1996*) and there is usually a positive correlation between leaves with a high LMA and drought tolerance (*Wright, Reich & Westoby, 2001*). However, if a severe drought does cause dieback in plants with high LMA, the greater carbon investment per leaf may require longer post-drought recovery times for construction of new leaves for those species than for species with low LMA (*Witkowski & Lamont, 1991*; *Wright et al., 2004*).

In 2011, Texas experienced the most severe drought in its recorded history (*Neilsen-Gammon, 2011*; *Combs, 2011*), which led to dramatic plant mortality in Big Bend National Park (*Poulos, 2014*; EF Waring & DW Schwilk, pers. obs., 2011). This drought was coupled with an unusual multi-day, severe freeze in February 2011, which likely exacerbated the effects of drought in addition to directly causing freezing damage (*Poulos, 2014*). We used this event to test how perennial plant response to extreme drought varied across elevations, plant growth forms and leaf traits. We asked the following questions: What was the relationship between elevation and stem dieback; and were any elevational patterns driven by turnover in functional groups or in leaf trait changes? To answer the second question, we investigated if susceptibility to drought differed among functional groups or by species leaf traits. These questions lead to two competing hypotheses regarding the interaction of drought and elevation. First, we would expect there to be higher proportional dieback at the lower elevational sites due to absolutely lower precipitation during the drought. However, the plants at the lower elevations may have greater physiological tolerances to drought conditions, so a second hypothesis is that we could see less proportional dieback at lower elevations, especially as the 2011 departure from average conditions was greater at higher, rather than lower, elevations.

## METHODS

### Site description

Plant communities at Big Bend National Park range from succulent and deciduous shrub dominated lowlands, to desert grasslands at mid elevations, to juniper, oak, and pine forest at higher elevations. Precipitation in Big Bend National Park is seasonally uneven

(*NOAA, 2013*). There is higher precipitation in the summer and fall months with very little precipitation in the winter and spring (*Robertson, Zak & Tissue, 2009*). After years of wetter than normal conditions, 2011 was an extraordinarily dry year for Big Bend National Park (*Neilsen-Gammon, 2011*). The average annual precipitation increases with elevation: at the Chisos Basin meteorological station (1,617 m above sea level) annual precipitation averaged 441.7 mm between 1947 and 2012, Panther Junction (1,143 m) averaged 333.04 mm between 1955 and 2012, and Rio Grande Village (566 m) averaged 167 mm annually in 2006 through 2012 (*NOAA, 2013*). In 2011, however, the Chisos Basin received only 109 mm of precipitation, the Panther Junction Station received 64 mm precipitation and Rio Grande Village received 59 mm (*NOAA, 2013*). This drought was coupled with extremely high temperatures in 2011 in Big Bend National Park. Prior to 2011, the Chisos Basin averaged 22 days with temperatures above 32 °C, while Panther Junction averaged 106 days, and Rio Grande Village averaged 154 days (*NOAA, 2013*). However in 2011, the Chisos Basin experienced 63 days, Panther Junction had 151 days, and the Rio Grande Village had 234 days above 32 °C (*NOAA, 2013*). In addition to the severe drought, there was a multi-day freeze event in February of 2011 that probably contributed to plant dieback (*NOAA, 2013*; *Poulos, 2014* for the Chisos Basin freeze).

We collected data at six elevations within the park (∼250 m apart vertically) in autumn of 2010 and 2011 (Table 1, conducted under permit BIBE-2010-SCI-0019 to DWS). The lower elevation sites were located along the Ross Maxwell Scenic Drive while the high elevation sites were along the Pinnacles Trail. Sites were selected to be near hiking trails and roads in 2010. Sites were chosen with a north to northwest aspect with the exception of the lowest elevation site which was nearly level. In each of the two sampling years, fifty-meter-long transects were placed randomly within a 500 m radius of the GPS location for each site. The six sites are described in Table 1. Species were identified using *Powell (1998)* and *Muller (1940)*.

## Canopy cover measurements

In 2010 and 2011, 4–12 fifty-meter-long line-intercept transects were run perpendicular to the slope at each site. The number of transects was dependent on the density of the cover at each site: sites with higher cover had fewer transects because we aimed for relatively even sampling effort at each site (number of transects per site in Table 1). Transect starting points were selected randomly each year. The intercepted distance of canopy cover was recorded for each individual woody plant. Overlapping canopies were measured individually leading to the possibility of having canopy cover >1. Herbaceous species were not identified to species but were grouped together and bare ground was recorded. In 2011, for each individual woody plant or succulent, the proportion of the total canopy that was dead was visually estimated and recorded as "dieback proportion" for each individual plant (tested by bending small twigs in the case that leaves were missing). Dieback data were not estimated in 2010 after preliminary observations indicated very little dieback (<2% of cover). We classified species according to growth form: tree, shrub, subshrub, or

**Table 1  Site descriptions.** A description of each collection site as well as information of the woody species that were present at each site.

| Elevation (m) | Location (latitude and longitude) | Number of transects | Dominant growth form | Most common species in transects | Aspect |
|---|---|---|---|---|---|
| 666 | 29°8.24′ N 103°30.8′ W | 12 | Deciduous shrub | *Larrea tridentata* | Flat |
| 871 | 29°10.8′ N 103°25.8′ W | 10 | Desert shrub | *Agave lechuguilla* *Jatropha dioica* *Larrea tridentata* | Flat |
| 1,132 | 29°21.2′ N 103°16.7′ W | 9 | Desert shrub | *Acacia greggii* *Agave lechuguilla* | Flat |
| 1,411 | 29°18.2′ N 103°15.9′ W | 7 (2010) 5 (2011) | Shrub | *Acacia constricta* *Dasylirion leiophyllum* *Opuntia chisosensis* | North |
| 1,690 | 29°16.0′ N 103°18.1 W | 4 (2010) 5 (2011) | Tree | *Juniperus* species (*J. coahuilensis, deppeana, flaccida,* and *pinchotii*) *Opuntia chisosensis* | Northwest |
| 1,920 | 29°15.2′ N 103°18.0′ W | 4 | Tree | *Juniperius deppeana* *Quercus species* (*Q. emoryii, gravesii,* and *grisea*) | Northwest |

succulent. We defined trees as species that regularly grew as single stemmed individuals >2 m tall; shrubs as species with mostly multi-stemmed individuals <2 m; subshrubs as individuals with some above ground woody stem and herbaceous growth above the base that would die back annually (e.g., equivalent to "chamaephyte" in Raunkiær classification, *Du Rietz, 1931*), and succulents included individuals of the families Agavaceae, Cactaceae, and Nolinaceae. Transects were treated as replicates nested within a site. All measurements were expressed on a per transect basis for analysis.

Total canopy cover for each individual was determined using Eq. (1).

$$C_i = \frac{\sum L_i}{50} \tag{1}$$

where $i$ represents the individual species, $L$ is the intercepting length of the canopy and 50 represents the length of the transect (50 m).

The total canopy cover for each transect was determined by Eq. (2).

$$C_t = \sum_{i=1}^{n} C_i \tag{2}$$

where the sum of all individual plant is determined for each transect. Relative canopy cover was determined using Eq. (3).

$$R_i = \frac{C_i}{C_t} \tag{3}$$

After calculating the total cover of each transect and relative cover, the amount of dieback per individual was calculated using Eq. (4).

$$D_i = \frac{\sum d_i L_i}{50} \qquad (4)$$

In Eq. (4), the nomenclature is the same as Eq. (1) and $d_i$ is the proportion of observed canopy dieback per individual. Using the dieback per individual, we were able to calculate total dieback (Eq. (5)) and total living canopy cover (Eq. (6)).

$$TD = \sum_{i=1}^{n} D_i \qquad (5)$$

$$C_l = \sum_{i=1}^{n} C_i - \sum_{i=1}^{n} D_i = C_t - TD \qquad (6)$$

Lastly, we calculated the amount of proportional dieback per transect using Eq. (7).

$$PD_i = \frac{D_i}{C_i} \qquad (7)$$

Proportional dieback for a single growth form was the total dieback distance of species in a particular growth form divided by the total cover of that growth form. Proportional dieback was used in all analyses of dieback.

In 2010 (pre-drought), we collected 3–8 leaves per individual for 2–3 individuals per species at each elevation. We did not collect leaves from succulents (Cactaceae, Nolinaceae and Agavaceae). We used a flatbed scanner and LAMINA software (*Bylesjö et al., 2008*) to calculate leaf area. After leaf area was recorded, the leaves were dried for 24 h at 85 °C and then weighed to determine the dry mass of the leaf. The LMA was calculated as dry mass over area (g/cm$^2$). The weighted LMA on each transect was the average of the species' LMA values on that transect weighted by each species' relative cover.

## Statistical analysis

All data were analyzed using R (*R Core Team, 2013*). All data were tested using a Shapiro–Wilk test of normality in R which confirmed that the response variables were normally distributed. Statistical differences across elevations were determined using ANCOVA models. For the analysis of total, living, and relative canopy cover, hierarchical linear models were run with elevation and year of data collection as fixed effects. Transects were nested within elevations: with one site per elevation, we have a sample size of six with which to detect elevational trends. Models were fit with the nlme package in R (*Pinheiro et al., 2013*). For models predicting relative canopy cover and dieback by elevation and growth form, transect was a random effect (due to multiple dieback estimates per transect) within elevation. The data for total and living cover were untransformed because they met the assumption of normality and were not true proportions. However prior to analysis, proportional dieback and relative cover were transformed using an empirical logit transformation ($\ln((p + \varepsilon)/(1 - (p + \varepsilon)))$, where $\varepsilon = 0.0001$) as those measurements were

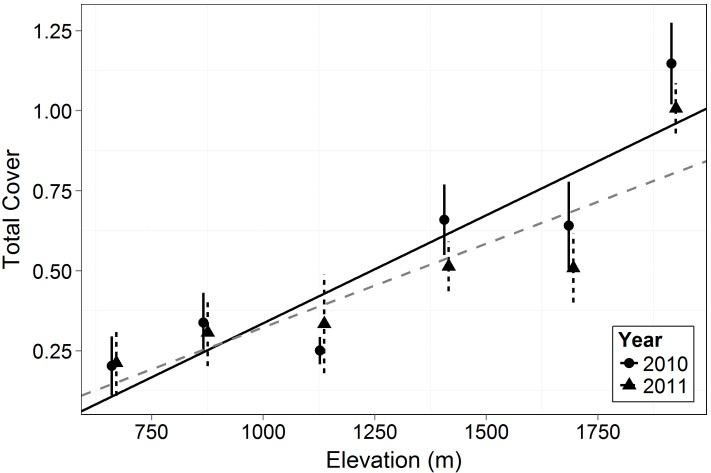

**Figure 1 Total canopy cover by elevation in 2010 and 2011.** Symbols represent the mean ±1 standard deviation. Circles and a solid line represent 2010 while triangles and a dashed line represent 2011. There was a significant increase in cover with elevation ($F = 17.34$, $p = 0.014$) and interaction in cover between years and elevation ($F = 6.895$, $p = 0.010$).

true proportions (*Warton & Hui, 2011*). For the analysis of wLMA by proportional dieback across elevations, a nested linear model was used (transects nested within elevation). Post-hoc analyses were done on the wLMA and proportional dieback including and excluding the conifer species. Additional analyses where the sites were grouped by high and low elevation were also performed.

## RESULTS

Total canopy cover (both living and dead canopy combined) for all growth forms increased significantly with elevation ($F = 17.34$, $p = 0.014$). In 2010 and 2011, the lowest elevation had ∼80% less cover than the highest (Fig. 1). Total canopy cover did not change significantly between years ($F = 1.549$, $p = 0.217$), but there was a significant interaction between elevation and year ($F = 6.895$, $p = 0.010$) with the highest elevations dropping slightly in total cover by the 2011 measurement (possibly a result of a slight underestimation of total cover for trees with some dieback and with leafless canopy that year). The relative cover of the four perennial woody growth forms (trees, shrub, subshrub, and succulent) differed significantly across elevation (empirical logit transformed, $F = 69.36$, $p < 0.0001$, Fig. 2). There was no significant difference in relative cover between years and no interaction between year and growth form. Succulent and shrub species were present at all elevations. Subshrubs were present at all elevations except the highest elevation, 1,920 m. Tree species were present at the two highest elevations, 1,920 and 1,690 m. Growth forms showed differing elevational patterns in relative cover (significant interaction between growth form and elevation, $F = 36.57$, $p < 0.0001$, Fig. 2).

Living canopy cover of all woody plants (shrubs, succulents, subshrubs and trees) increased with elevation ($F = 18.33$, $p = 0.013$, Fig. 3). Living canopy cover decreased significantly post-drought ($F = 103.6$, $p < 0.0001$) and there was no interaction with elevation: the absolute amount of dieback was consistent across elevations. Because

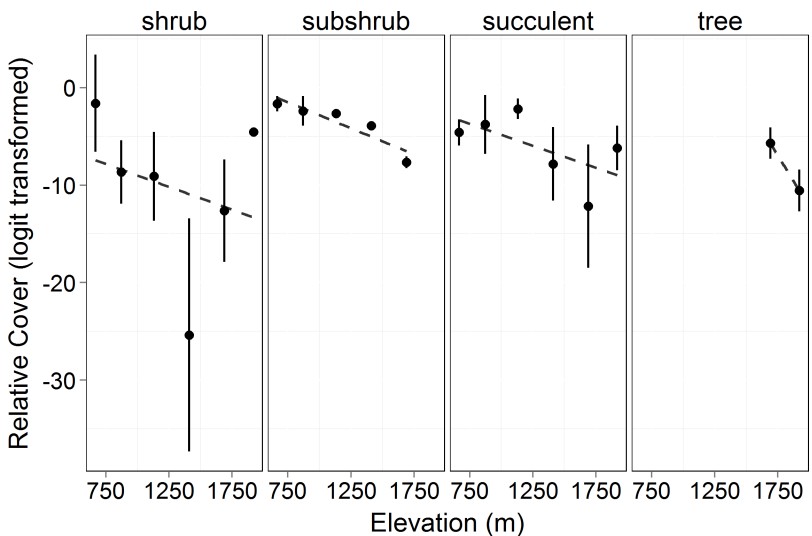

**Figure 2 Relative cover (logit transformed) of different growth forms by elevation in 2010.** This shows the cover of each growth form relative to the total canopy cover (see Eq. (3)) and the changes in growth forms on an elevational gradient. Circles represent the mean ±1 standard deviation. Only 2010 is shown in the figure as there was no significant difference between years ($F = 3.473$, $p = 0.064$). There was no significant difference among the growth form cover across elevations ($F = 0.0466$, $p = 0.840$). However, there was a significant difference in the relative cover of the growth forms ($F = 69.36$, $p < 0.0001$) and in the interaction of growth form with elevation ($F = 36.57$, $p < 0.0001$).

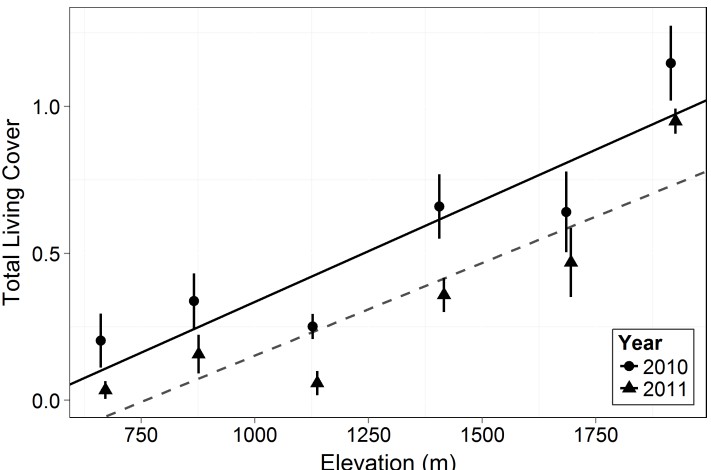

**Figure 3 Total living canopy cover by elevation in 2010 and 2011.** Symbols represent the mean ±1 standard deviation. Circles and a solid line represent 2010 while triangles and a dashed line represent 2011. Living cover increased significantly with elevation ($F = 18.33$, $p = 0.013$) and there was a significant decrease in 2011 compared to 2010 across elevations ($F = 103.6$, $p < 0.0001$).

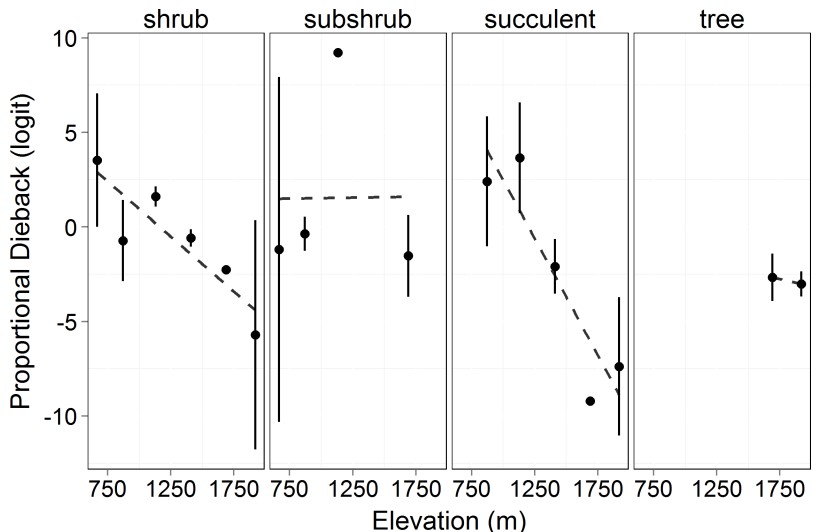

**Figure 4 Proportional canopy dieback (logit transformed) in 2011 by elevation.** Circles represent the mean ±1 standard deviation. This shows the amount of dieback at each elevation in proportion to the amount of total cover measured (see Eq. (7)). Transformed proportional dieback decreased with elevation across all growth forms ($F = 8.867$, $p = 0.04$). There was no significant difference in proportional dieback among growth forms ($F = 0.844$, $p = 0.473$), but there was an interaction between growth form and elevation ($F = 7.245$, $p = 0.0002$).

**Table 2 Proportional dieback of growth forms by elevation.** Each value represents the untransformed mean proportional dieback of each growth form across elevations ±1 standard deviation. The bottom row is the total mean of proportional dieback for all growth forms at each elevation. If a growth form was not measured at an elevation, it is represented with NA.

|  | 1,920 m | 1,690 m | 1,411 m | 1,132 m | 871 m | 666 m |
|---|---|---|---|---|---|---|
| Shrub | 0.26 ± 0.45 | 0.10 ± 0.02 | 0.36 ± 0.11 | 0.82 ± 0.08 | 0.35 ± 0.28 | 0.85 ± 0.12 |
| Subshrub | NA | 0.29 ± 0.20 | NA | 1.00 ± 0 | 0.42 ± 0.19 | 0.42 ± 0.49 |
| Succulent | 0.03 ± 0.07 | 0 ± 0 | 0.17 ± 0.14 | 0.90 ± 0.11 | 0.74 ± 0.32 | NA |
| Tree | 0.05 ± 0.03 | 0.10 ± 0.07 | NA | NA | NA | NA |
| **Total** | **0.10 ± 0.23** | **0.11 ± 0.14** | **0.27 ± 0.16** | **0.88 ± 0.11** | **0.50 ± 0.31** | **0.68 ± 0.38** |

high elevations had much higher initial cover, however, this resulted in much lower proportional dieback at the higher elevations: living canopy cover decreased by 17% at the highest elevation, 1,920 m, and by 83% at the lowest elevation, 666 m.

Proportion dieback decreased significantly with elevation in succulent and shrub species (logit transformed, $F = 8.867$, $p = 0.04$). There was no effect of growth form on proportional dieback ($F = 0.844$, $p = 0.473$), but there was an interaction between growth forms and elevation ($F = 7.245$, $p = 0.0002$, Fig. 4). Proportional dieback for each site varied from 0.90 (succulents at 1,132 m) to 0.03 (succulents at 1,920 m) (Untransformed means, Table 2). The overall elevational trend was driven by succulent and shrub species for which dieback decreased with elevation (Fig. 4).

**Peer**J

There was no relationship between wLMA and dieback across transects within elevations (nested ANCOVA, $F = 2.096$, $p = 0.152$). Because conifer and angiosperm leaves differ in gross morphology, we also conducted a similar nested ANCOVA which excluded conifers. Those results were consistent with the analysis including conifers: there was no relationship between wLMA and dieback ($F = 0.034$, $p = 0.855$) nor was there an interaction wLMA and elevation ($F = 0.550$, $p = 0.463$). We also grouped sites into two elevation classes and ran the nested ANCOVA with elevation class ("high" = 1,920, 1,690, and 1,411 m and "low" = 1,132, 841, 666 m) as a factor. There was no relationship between wLMA and dieback ($F = 0.992$, $p = 0.326$) nor was there an interaction between wLMA and elevation class when conifers were included ($F = 0.017$, $p = 0.898$) or excluded ($F = 0.088$, $p = 0.768$).

## DISCUSSION

As in most arid mountain ranges, total woody plant cover increased with elevation (*Whittaker & Niering, 1964*; *Whittaker & Niering, 1965*; Fig. 1). The lack of significant difference in total canopy cover across years indicates that our sampling methods were consistent across years despite the large amount of dead canopy at some elevations in 2011—with the possible exception that we may have slightly underestimated total cover in 2011 at the highest elevation site (Fig. 1). The change in canopy cover from dominantly trees at higher elevations to shrubs at lower elevations was the expected community turnover in Big Bend National Park (Fig. 2). Living canopy cover was affected by the drought of 2011 and living cover was significantly reduced in 2011 at all elevations, as would be expected (Fig. 3). However, the large degree to which living cover decreased in 2011 (largest decrease at 1,411 m from 0.65 to 0.35) was striking. We expected the extreme drought event to overwhelm some of the adaptations that the lower elevation species had for drought. However, as the dieback data indicates, this drought affected all growth forms at all elevations. The impacts of drought across growth forms resulted in the relative abundance of each growth form at a given elevation (measured by living cover) not changing as a result of the drought. Because most of our sites were on a north to northwest facing aspect with less solar radiation than southern aspects, it is likely that the dieback measured in this study is a conservative estimate of the amount of dieback that occurred in Big Bend National Park in 2011.

By the end of 2011, the Chisos Basin (1,615 m), the Panther Junction (1,140 m), and the Rio Grande Village (566 m) meteorological stations had only received about 22%, 21%, and 39% of their normal average precipitation at each location annually (*NOAA, 2013*). This decrease in available water would have increased the likelihood that all woody species experienced stress in xylem transport potentially leading to hydraulic failure. *Kukowski, Schwinning & Schwartz (2012)* reported extensive tree dieback due to hydraulic failure in the Edwards Plateau of Central Texas after the 2011 drought. Our study showed total dieback did decrease significantly with elevation and increasing precipitation. Additionally, this drought was coupled with an unusual multi-day freezing event in February 2011 (*Poulos, 2014*; *NOAA, 2013*). The freeze in 2011 affected the various elevations differently

**Table 3 Freezing days and minimum daily temperature in November 2010 to March 2011.** The number of freezing temperature days and minimum daily temperature at each meteorological station in the winter of 2010/2011 compared with yearly averages (in parentheses). The Chisos Basin meteorological station is located slightly lower in elevation than our second highest site at 1690 m. The Panther Junction meteorological station is located at a similar elevation to our first Chihuahuan Desert site at 1132 m and the Rio Grande Village meteorological station is close to our lowest elevational site (666 m), but is located closer to the river and slightly down stream. Zero degree days are any day where the maximum temperature for that day does not get above 0 °C. Freezing days are days where the minimum daily temperature was below 0 °C. Across the meteorological stations, February 2011 was much colder than normal while the other months were generally warmer than average.

| | Chisos Basin (1,615 m) | | | Panther Junction (1,140 m) | | | Rio Grande Village (566 m) | | |
|---|---|---|---|---|---|---|---|---|---|
| | Average minimum temperature (°C) | Zero degree days | Freezing days | Average minimum temperature (°C) | Zero degree days | Freezing days | Average minimum temperature (°C) | Zero degree days | Freezing days |
| Nov | 7.2 (6.2) | 0 (0.45) | 1 (9.0) | 7.4 (6.8) | 0 (0.11) | 1 (2.6) | 3.5 (4.0) | 0 (0) | 3 (5.4) |
| Dec | 4.7 (3.4) | 0 (0.26) | 4 (5.9) | 4.4 (3.0) | 0 (0.19) | 4 (7.8) | −1.9 (−0.6) | 0 (0) | 21 (17.8) |
| Jan | 3.3 (2.8) | 0 (0.05) | 6 (3.0) | 2.6 (2.3) | 0 (0.46) | 10 (10.0) | −1.5 (−1.2) | 0 (0.2) | 21 (20.6) |
| Feb | 3.3 (4.0) | 3 (0.11) | 10 (3.7) | 2.4 (4.1) | 2 (0.23) | 10 (5.7) | −1.8 (−1.2) | 2 (0.5) | 15 (11) |
| Mar | 12.0 (6.8) | 0 (0.31) | 0 (8.4) | 11.1 (7.6) | 0 (0) | 0 (2.2) | 8.5 (6.0) | 0 (0) | 0 (3.8) |
| **Total** | **6.1 (4.6)** | **3 (1.18)** | **21 (30)** | **5.6 (4.8)** | **2 (0.99)** | **21 (29.29)** | **1.4 (1.4)** | **2 (0.7)** | **60 (58.6)** |

(Table 3). The interaction of freezing and drought may have exacerbated hydraulic failure (*Langan, Ewers & Davis, 1997*; *Cavender-Bares & Holbrook, 2001*; *Martinez-Vilalta & Pockman, 2002*). It is likely that the freeze also contributed to the extensive succulent mortality we witnessed, but there is not an obvious difference in the departure from average across the elevations: at all three meteorological stations, there were about 1.5 times as many freezing days as normal in February 2011 (Table 3).

Although the greatest reduction in precipitation and the most prolonged freezing temperatures were at the highest elevations, the effect of the drought (and possibly freezing) were most severe at the lower elevations in terms of proportional canopy dieback (*NOAA, 2013*). Therefore, it seems likely that the more severe effects at low elevation are due to species at those sites being closer to physiological thresholds. This is consistent with our first hypothesis that the lower elevations would experience more dieback due to lower precipitation and/or soil moisture. One site (871 m) defied the trend: this site had less dieback than the next higher site (1,132 m, Fig. 4). This was most likely due to the topography of the site at 871 m. That site was characterized by a gentle slope, but in a drainage that could have led to extra water being funneled to the site through runoff.

Overall, the degree to which the drought affected the different growth forms varied with elevation (Fig. 4). The overall trend was decreasing dieback with elevation, but this was driven by the dominant shrub and succulent growth forms. Trees only occurred at two elevations and we therefore cannot test an elevational effect. Tree dieback levels were intermediate in comparison with the range of values seen for shrubs across elevations (Fig. 4). There was no effect of elevation on dieback in subshrub species. For subshrub species, the lack of differences in dieback was due to where they were located and their responses to drought. There were very few subshrubs at 1,690 m, so their dieback would

have little influence on the overall dieback for the subshrubs. Additionally, subshrubs have a woody base and herbaceous annual growth. We hypothesize that there was little canopy dieback in subshrubs because, while the woody stems persisted, the herbaceous growth on the subshrubs never leafed out in 2011 due to the drought. The previous year's growth dried and abscised prior to our measurements, therefore there was less cover to measure in 2011. The drought did not affect subshrubs through canopy dieback, but rather through preventing growth.

The shrubs were greatly affected by the drought at all elevations lower than 1,690 m (Fig. 4). Plants exhibit different strategies to deal with drought including variation in rooting depth, variation in ability to prevent water loss, and differences in tissue-specific drought resistance (*Schwinning & Ehleringer, 2001*; *Chesson et al., 2004*; *Ogle & Reynolds, 2004*). Despite this, we saw significant mortality across both evergreen and deciduous species. Significant change in total living cover across elevations was driven by the magnitude of dieback in shrub species (Fig. 4). The succulent species were expected to be able to tolerate the drought better than the other growth forms as a result of water efficient CAM photosynthesis and large reservoirs of water stored in their stems. However, even for succulents, there was significant dieback at the lower elevations and little dieback at the higher elevations. The severity of the drought (or the combination of drought and freezing) at Big Bend National Park was too stressful for the succulents at the lower elevations.

The relationship between elevation and dieback is not explained by differences in wLMA among species for which leaf traits were collected. Leaf traits were only collected on shrub, subshrub, and tree species. There was no relationship between LMA and dieback on a species or growth form basis. The species at the lower elevations are adapted to drought by producing small, easily replaceable leaves, and while investment per leaf area was generally higher at higher elevations, there was no effect of wLMA on dieback within elevations (data not shown). This does not mean that growth form and wLMA were not factors in tolerance to the drought, but with our data, elevation alone is the strongest predictor of the impact of the 2011 drought and freeze event. Despite the desert species in the lower elevations having adaptations for drought tolerance, the historic severity of the 2011 drought overpowered those adaptations which led to higher dieback in drought tolerant species.

## CONCLUSION

The 2011 drought in Big Bend National Park had a large impact on all plant communities, with the relative effects decreasing with elevation. Our data imply that differences in canopy cover and dieback among elevations were probably driven by differences in absolute drought and freezing severity and not by turnover in growth forms and not by turnover in growth forms, or leaf trait differences. Species turnover within growth forms (e.g., variation in drought tolerance within the diverse shrub group) may have played a role in the elevational trend, but we cannot test that with our data. We can say, however, that the decreasing dieback with elevation was not driven by shifts in the relative abundance

of shrubs, succulents and trees. Nor was an easily measurable leaf trait, LMA, driving the observed trend.

We cannot distinguish the relative influence of drought and freezing on these communities, but we suspect that freezing may be an important environmental factor in this system. One possibility is that the freezing event most severely impacted low elevation species, and the resultant low elevation mortality masked differences in drought susceptibility (although meteorological data suggest that, historically, winter freezes are just as common at low elevations as they are at the higher). We believe the most likely explanation is that this extreme drought pushed all species to the edge of their tolerances. The widespread dieback of slow-growing shrub and succulent species in the lower elevations of Big Bend National Park will likely have long-term effects on the plant community.

## ACKNOWLEDGEMENTS

We thank Joe Sirotnak at Big Bend National Park for his assistance and we thank Brandon Pratt and the students in Ecological Strategies of Plants; Will Brewer, Tony Cullen, Shirali Davé, Thomas Fitzgibbons, Maria Gaetani, Nathan Geiger, Hasitha Guvvala, Audrey Harrell, and Caleb Hill for their contributions to data collection. We thank Michael Huston, Rita Quiñones-Magalhães, Susanne Schwinning and an anonymous reviewer for their comments which improved this manuscript.

### Funding

This work was funded by general teaching funds provided to the Ecological Strategies of Plants course by the Department of Biological Sciences at Texas Tech University. The funders had no role in study design, data collection and analysis, decision to publish, or preparation of the manuscript.

### Competing Interests

The authors declare there are no competing interests.

### Author Contributions

- Elizabeth F. Waring performed the experiments, analyzed the data, contributed reagents/materials/analysis tools, wrote the paper, prepared figures and/or tables, reviewed drafts of the paper.
- Dylan W. Schwilk conceived and designed the experiments, performed the experiments, analyzed the data, contributed reagents/materials/analysis tools, wrote the paper, prepared figures and/or tables, reviewed drafts of the paper.

### Field Study Permissions

The following information was supplied relating to field study approvals (i.e., approving body and any reference numbers):

This work was conducted under National Park Service permit BIBE-2010-SCI-0019 to Dylan Schwilk.

## Data Deposition

The following information was supplied regarding the deposition of related data:
    Schwilk lab GitHub repository.

## Supplemental Information

Supplemental information for this article can be found online at http://dx.doi.org/10.7717/peerj.477.

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
