# Peer review of "Plant dieback under exceptional drought driven by elevation, not by plant traits, in Big Bend National Park, Texas, USA"

_PeerJ, doi:10.7717/peerj.477_

## Round 0.1 · original submission · Major Revisions

Both reviewers agree that this is an important topic, and potentially a useful contribution to understanding drought mortality patterns in this system. However, both identify a number of issues - in clarity, statistics, and interpretation that must be addressed if the paper is to be published. I look forward to seeing a revision that addresses these issues.

Reviewer 1 ·

Basic reporting

The idea of the study is interesting. It would be nice to know more about recent droughts and their effects on the plants of Big Bend. However, that information is difficult to get from this ms. It is not clear what the authors say they have done. What they did doesn’t seem to matches what they say they did or imply they did and the material they present. There are a number of typos and thing like that in the text that should have been caught in proofing.
Figure 1-Total cover is not explained very well and it appears to be relative cover. Cover per se doesn’t really mean much, but presented as a percent helps put it in perspective. The analysis of the data is not presented very clearly in the methods or the figure. An ANCOVA is fine, but an f value would help with dfs. But that is general, where are the specific differences? Was a multiple range test used? Which one? Why not presented? The lines on the figure suggest or imply a regression but nothing is said or given about regressions. As such no lines should be presented and possibly a bar graph would be more appropriate, one showing where significant differences are.
Figure 2-Same comments here. Are cover values presented and the transformed data analyzed? This is hard to tell. The lines seem to be parallel, thus the slope would be the same, but if the lines are just eye-balled, it doesn’t matter. What do the transformed data lines look like? What is the r, R2, and the equation for the line? What does the residual plot look like? What information can be obtained from these types of plots?

Experimental design

There isn’t any indication if the data presented meets the requirements to do parametric statistics. Are distributions normal? Are variations equal? Was this requirement examined or tested? What test was used?
The data treatment in the methods is very brief and too short to tell what was done with the data.
Looks like solid-filled circles are used in both figures 3 and 4, but the legends say something else. Straight lines are used throughout, but there is no indication that these relationships are linear.

Validity of the findings

An ANCOVA is fine, but an f value would help with dfs. But that is general, where are the specific differences? Was a multiple range test used? Which one? Why not presented? The lines on the figure suggest or imply a regression but nothing is said or given about regressions. As such no lines should be presented and possibly a bar graph would be more appropriate, one showing where significant differences are.
What do the transformed data lines look like? What is the r, R2, and the equation for the line? What does the residual plot look like? What information can be obtained from these types of plots?
The data treatment in the methods is very brief and too short to tell what was done with the data. Thus, it is very difficult to draw and valadate conclusions.

·

Basic reporting

The report fails in clarity in several categories: data sampling methods, analysis and results.

Experimental design

Seems ok

Validity of the findings

The findings are valid, although the conclusion is overstated by suggesting cause and effect.

Additional comments

I think these data are very valuable, but that the report could still be improved if more information was provided about elevation trends in several potential drivers of stem mortality, not just precipitation, but also temperature both at the high end, and the low end, since a freezing spell was mentioned in the report that also could have affected mortality.

The authors provide information on precipitation and its % deviation from average, but the one thing that made the 2011 drought truly outstanding was temperature, particularly in the – still rainless – month of September. The authors mention as a further potential cause of mortality freezing temperatures. So I would suggest as Fig. 1 an overview of elevation trends in mean precip (e.g., March to September), mean cumulative degree days and maximal numbers of consecutive days below zero centigrade, as well as the % deviation of these values from the mean in 2011.
Additionally, what I would find interesting is a frequency distribution of %dieback, i.e. what % of individuals died outright and which only partially died back?

In general, the manuscript needs a fair bit of clean-up, as indicated by my detailed comments below.

Line 13: which productivity-diversity relationship? – reference?
Line 19: is there a reference for that last assertion? – Pockman and Sperry comes to mind.
Lines 47-51: These additional hypotheses are distracting, as they describe well-known landscape patterns. These could perhaps be replaced with a hypothesis about trends with vegetation type, which you also test. E.g., mortality rates being unevenly distributed among plant functional types: sub-shrubs>trees>shrubs>succulents - maybe.
Line 72: lower case in fifty-meter-long.
Line 76: avoid repetition.
Line 80: intercepted distance or greatest canopy width? These should be different. How did you handle overlapping canopies? Seems to me there was a fair bit of overlap because the relative cover of different plant types adds to > 1.
Line 94: the calculation of living canopy cover needs some clarification. You measured intercept lengths and estimated %dieback per individual, so you should multiply individual dieback fractions with intercept lengths and then add these up and subtract from total cover. Is that what you did? It may be easier to just show the formula used.
Line 103-105: this sentence needs fixing.
Lines 106-112: In the ANCOVA, which factor is fixed and which covariate? If transects were “nested” what was the sampling unit?
Line 128: Reference to Figure 4 at the end of this sentence. Also, this sentence contradicts the legend to Figure 4. Or perhaps you wanted to say that “averaged across all growth forms, dieback decreased ….”
Line 132: Fig. 4 is referenced before Fig. 3 in the next paragraph.
Lines 133-138: since this does not address the main gist of the paper, perhaps show this earlier as helping to characterize the sites. Also, the following paragraph repeats what is said in this paragraph, except it was done for both years, after you explained in the legend that relative cover did not change between years. This is confusing.
Fig. 5: Why are there so many points? You explain in the methods (I think) that wLMA was calculated as the species- weighted average by elevation. So I would have expected one point per elevation. Or is this one point per species per elevation? – If so, I would also check if there are significant correlations within elevations, because stress intensity was different at different elevations. Actually, this analysis calls for an ANCOVA with elevation as fixed effect and wLMA as covariate.
Line 175: For that reason, it would be interesting to know how the freeze expressed at different elevations and how different that was from average.
Line 184: I don’t understand why you expected relative cover to change, since you counted dead and live cover, so I don’t think it indicates anything except that dead biomass does not immediately disappear. Also, Fig. 3 does not support this assertion. I think you wanted to refer to Fig. 2 and live cover. I would also disagree with this conclusion, since different cover types could have had different dieback proportions and still produce the pattern in Fig. 2. Figure 4 does in fact indicate that different types are differentially susceptible to drought.
Line 190: Not clear what you mean by “herbaceous, annual growth” at the base of the sub-shrubs. The way you explained it, the %dieback was taken to be 0 in 2010 across species and elevations. So if sub-shrubs did not leaf out in 2011, they should have been recorded as 100% dead in 2011. This needs to be better explained
Line 205: So what is your conclusion regarding your question whether mortality patterns were driven by species turnover? Was relative dieback greater at lower elevation because there were more shrubs and shrubs had stronger responses to drought intensity (or the freezing spell)? An informative analysis you could do in this regard is to show what %of total dead cover was accounted for by each veg type across elevations and contrast this with the relative abundances of veg types. The Null hypothesis is that each veg type contributes a fractional mortality that is proportionally to its abundance.
Line 207: you don’t know this. But you can say that variation in LMA did not explain trends of mortality with elevation.
Lines 180-220: You do not have any evidence for this assertion. In fact, in terms of % below average, lower elevations were less affected. Especially since there is an additional weather trend that you did not examine as much as precipitation: the freeze. Also, I am not convinced that you can exclude species turnover as part of the explanation: there were more trees at higher elevation and trees were also less susceptible to drought.

---

## Round 0.2 · Major Revisions

Editorial&Review comments on “Plant Dieback under exceptional drought... Waring and Schwilk” Revision

While the authors did address many of the issues raised by the reviewers, the submitted revision was so sloppy and full of errors that it would be an insult to send it out to reviewers again. It is obvious that the manuscript had not even been proofread, with numerous missing words, misspellings, and other errors.

There is still a major lack of clarity in how the main variables were calculated from the field data, exemplified by the incomplete and /or misprinted equation at line 98. One of the reviewers has provided a set of equations for all the major calculations, based on assumptions about what the authors were trying to calculate. This will be sent separately,

I am willing to consider one more revision, but the submitted manuscript must be perfect, with no typos, missing or misspelled words, or other sloppy errors.

Errors in each of the following lines must be corrected. There are probably others that I missed. I should not have to waste my time with a manuscript into which the authors put so little effort.
lines 12, 29, 64, 105, 150, 154, 169,

line 31 replace “ever recorded” which is not true and replace with “in its history”
line 97.5 rewrite equation with simple, defined variables and appropriate symbols and clearly defined variables.
line 100-104 “dieback length” “cover length”
line 116 “... normality in R which confirmed that all the data were...”
line 127 replace subtracted with added? correct?
line 129 0 or 0.00001 ?
line 157,158 use percentage rather than proportion, most of your discussion and data are presented as percentage changes.
lines 162 – 165 – consider deleting all material about LMA. The high elev conifers have a completely different leaf structure than the broad-leafed species and are clearly LESS drought toleraant or they would also grow at lower elevations. Fig. 5 shows two contrasting groups of sites, both with a surprising increase in dieback with increasing wLMA. There is one extreme outlier from 1132m. What;s going on here? You should either completely rethink and redo your analyses of LMA. There may be something interesting here, but you have completely missed it.
line 174 “at lower elevations reflected the known community turnover...”
line 175, 177 “Total ??living?? cover”
line 180-182 – Figure 2 is irrelevant to this sentence
line 191 “decrease significantly with elevation and increasing moisture”
line 192 NOBODY says “the first hypothesis we posited”
change to “in line with our first hypothesis.”

lines 194 - 208 add specific values for temperatures during freeze, NOT just the average minimum for the month. the data you present should be relevant to the event. What were the lowest temperatures? What is the normal number of freezing days in each month, etc.


lines 247 – 259 the LMA and dieback data need to be completely re-thought, re-analyzed, and re-written. Fig. 5 needs to be redrawn and re-analyzed. You have a clear contrast between the lower 3 sites and higher three sites. The sentence beginning “The higher abundance of drought tolerant trees at the higher elevations .... “ simply does not make sense. LMA is not necessarily a determinant of drought tolerance in all cases.

Table 1 – give number of transects at each elevation.

Fig. 1. Explain sample size numbers in F statistics – 4? 83? what?

Fig. 2. Logit ?regression?? what? explain sample size numbers. Relative cover is relative to what? this figure is difficult to understand. Interaction of each growth form with what??

Fig. 3. typo in legend. can you find it?

Fig. 4 same questions as Fig. 2

Fig. 5 Rethink and redo, separating upper vs lower 3 elevations.

---

## Round 0.3 · Minor Revisions

This manuscript continues to improve and should be ready for publication with a few more clarifications.

Abstract

Wording changed in the main text should also be changed in the abstract.
change first line to “... the most severe single year drought in its recorded history, resulting....”
fourth from last line... “differences in precipitation and drought severity” is redundant.
change to “...differences in drought severity.”

line 20 “any relationship” is too vague and “?;” is inappropriate. Change this to “stem dieback? Were any elevational patterns driven by turnover...” if this is what you were trying to express.

line 46-47 Note that line 43 introduces a “First” to which there is no obvious “second.” change to “... tolerances to drought conditions, so a second hypothesis is that we could see less proportional dieback at lower elevations, especially as....”

line 54. no semicolon is needed. End sentence and start another: “There is higher...

line 80. text would be easier to understand as: “In 2010 and 2011, from 4 to 12 fifty-meter-long line-intercept...”

line 87 after mention of herbaceous plants, change following to “.. for each individual woody plant or succulent...”

line 95 add space between comma and “du Rietz...”

line 107 I believe that “percent” should be changed to “proportion” if the formula is to work correctly.

line 164 You should make it completely clear what “proportional dieback” applies to. Is it per site, per transect, or per growth form? Eq 7 indicates it is by growth form, but the general concept of “proportional dieback” could apply to transects or sites as well.

In line 158 you use the term “proportional dieback” with obvious reference to total vegetation, combining all growth forms.

However, in the next paragraph you discuss proportional dieback by growth form, and refer to Fig 4 which shows proportional dieback by elevation by growth form. You state that “proportional dieback varied from 0.83 at 1132 M to 0.08...” This 0.83 is the same as the 83% mentioned for total vegetation in the previous paragraph, even though the focus of this paragraph is the growth forms. You must be clearer about your results.
One possibility would be to mention the highest PD by growth form in this paragraph, e.g. “Proportional dieback per growth from varied from 0.98 (shrubs at xxxx m) to 0.02 (trees at xxxx m) – or whatever it is.

I suggest modifying line 161 as “Proportion dieback decreased significantly with elevation in two of the four growth forms (logit tranformed, F....., Fig. 4).” Otherwise, you are simply repeating the same information about total proportional dieback that you provided in the previous paragraph.

This clarification is essential because subsequent text (line 186) says that the maximum decrease in living cover was “as much as 30% decrease at some sites,” which seems to be another expression of “proportional dieback.” Perhaps I don’t understand the distinction between “proportional dieback” (expressed as proportion) and “decrease in living cover” (expressed as percentage). This needs to be clarified.

While it is clear that logit (or other) transformation of proportions are essential for statistical analyses, I do not believe that your results will be useful or interpretable to most readers unless you present your basic data untransformed in additional tables.

You need two additional tables showing the drought impacts by elevation:

Table 3 (or renumber them and make this Table 2)
Proportional Dieback by Growth form (columns or rows) by elevation/site (rows or columns).
This should include the mean and variation (SD) in proportion based on the transects. Since no comparative statistics are applied to these data, this should not be a problem)
There should also be another column (the first) that shows total proportional dieback at each site (elevation)
This is important because the logit transformed values in Fig. 4 are difficult to relate to actual values of proportional dieback

Table 4 (or whatever)
Proportional dieback by species (ordered within growth forms) by elevation/site
Totals by species per site rather than mean & SD by transect would probably be appropriate – use your own judgment

And while we’re on tables...

Table 2 (on the cold weather – could end up as Table 4) needs another piece of information that every reader will wonder about and you can obviously provide
There should be an additional column for each station that shows the Monthly Minimum Temperature (with the yearly average monthly minima). Also, specify the time period over which your yearly averages were calculated.


line 181-2 “The change in canopy cover from....”

line 184-5 “...at all elevations, as would be expected (Fig 3). However, the large degree to which living cover....”

line 198 “... in xylem transport potentially leading to ....”

line 209 the 2011 precip in Chisos basin was 25% of average, at Rio Grande village it was 35% of average.
You might want to begin this paragraph with “Although the greatest reduction in precipitation and the most prolonged freezing temperatures were at the highest elevations, the effect of the drought (and possibly freezing) were most severe at the lower elevations in terms of....” Follow this with the sentence “Therefore, it seems likely....” followed by the sentence “This is consistent with our first hypothesis...” and DELETE the sentence “On the other hand,.... from average.”

line 222 “... cannot test an elevational effect. Tree dieback levels were ....”

line 253 You have presented NO EVIDENCE for the conclusion that “The higher abundance of drought tolerant trees at the higher elevations could have influenced the trends.....” There is no evidence that these trees are more drought tolerant than the shrubs at lower elevations, since the trees grow in an area that is cooler and received more precipitation than the lower elevations where the shrubs grow. If the trees were drought tolerant, they would be growing at lower, drier elevations. This is as misguided as your previous LMA conclusions. DELETE THIS SENTENCE.

line 260 “... with the relative effects decreasing with elevation.” You’ve made the point that absolute effects did not change with elevation, so this should be specific in the conclusion.

line 262 “... and not by turnover in growth forms, or leaf trait differences.”

line 274 hyphenate “slow-growing” and 276 “long-term”


I think this paper should attract a lot of interest once these final details are worked out.

---

## Round 0.4 · accepted · Accept

Your (and my!) persistence have finally produced a publishable manuscript that everyone concerned about drought impacts should find very interesting. I hope that these results, along with the work by Helen Poulos, will get the attention they deserve. Good work!